# Paper-Based Probes with Visual Response to Vapors from Nitroaromatic Explosives: Polyfluorenes and Tertiary Amines

**DOI:** 10.3390/molecules27092900

**Published:** 2022-05-02

**Authors:** Roberto Aguado, A. Rita M. G. Santos, Saúl Vallejos, Artur J. M. Valente

**Affiliations:** 1CQC, Department of Chemistry, University of Coimbra, Rua Larga, 3004-535 Coimbra, Portugal; roberto.aguado@udg.edu (R.A.); anaritasantos7890@gmail.com (A.R.M.G.S.); svallejos@ubu.es (S.V.); 2LEPAMAP-PRODIS Research Group, University of Girona, M. Aurèlia Capmany 61, 17003 Girona, Spain; 3Departamento de Química, Facultad de Ciencias, Universidad de Burgos, Plaza de Misael Bañuelos s/n, 09001 Burgos, Spain

**Keywords:** cellulose, explosive detection, Meisenheimer complex, nitroaromatics, paper analytical devices, polyfluorenes

## Abstract

Although it is well-known that nitroaromatic compounds quench the fluorescence of different conjugated polymers and form colored Meisenheimer complexes with proper nucleophiles, the potential of paper as a substrate for those macromolecules can be further developed. This work undertakes this task, impregnating paper strips with a fluorene-phenylene copolymer with quaternary ammonium groups, a bisfluorene-based cationic polyelectrolyte, and poly(2-(dimethylamino)ethyl methacrylate) (polyDMAEMA). Cationic groups make the aforementioned polyfluorenes attachable to paper, whose surface possesses a slightly negative charge and avoid interference from cationic quenchers. While conjugated polymers had their fluorescence quenched with nitroaromatic vapors in a non-selective way, polyDMAEMA-coated papers had a visual response that was selective to 2,4,6-trinitrotoluene (TNT), and that could be easily identified, and even quantified, under natural light. Far from implying that polyfluorenes should be ruled out, it must be taken into account that TNT-filled mines emit vapors from 2,4-dinitrotoluene (DNT) and dinitrobenzene isomers, which are more volatile than TNT itself. Atmospheres with only 790 ppbv TNT or 277 ppbv DNT were enough to trigger a distinguishable response, although the requirement for certain exposure times is an important limitation.

## 1. Introduction

Along this century, researchers have answered the call for facile detection of explosive nitroaromatic compounds in productive and scientifically sound ways, usually taking advantage of the electron-withdrawing capabilities of the −NO_2_ group [1,2,3]. Reasons include the prevention of terrorist attacks, the inspection after they have taken place, the assessment of water pollution or soil contamination by explosive manufacturing sites, the safety of operators during demining, and improving the accessibility of explosive trace detection systems [4,5,6]. The response is often based on a fluorescence turn-off mechanism [1,6,7], that phenomenon being mediated (in most cases) by photoinduced electron transfer (PET), resonance energy transfer (RET), or by both [8]. In parallel, the visual detection by the formation of a colored complex between a nitroarene and a nucleophile can also be found in the literature [3,9,10]. It should be clarified, nonetheless, that direct detection by these mechanisms is not aimed to compete with computed tomography, X-ray imaging systems or mass spectrometry, in terms of sensitivity, but to offer an easy and user-friendly approach wherever advanced techniques and qualified staff are not available.

It is worth exploring the two aforementioned possibilities, fluorescence quenching (requiring, at most, cheap UVA lamps) and color change under visible light, on a disposable substrate, manufactured from renewable resources, and highly available—cellulose paper [11]. There have already been valuable advances in this direction. For example, in a highly cited work, Ma et al. reported the impregnation of filter paper with 8-hydroxyquinoline aluminum (Alq3)-based nanospheres, and their fluorescence was quenched by 2,4,6-trinitrotoluene (TNT) solutions [12]. In an approach that is more similar to the one presented here, filter paper strips were drop-coated with polytriphenylamine derivatives that were effectively quenched by nitroaromatics in the micromolar range [13]. As for the colorimetric approach, polyaniline-coated papers have also been found to turn reddish in the presence of TNT, forming a Meisenheimer complex, with visually noticeable changes even when the analyte concentration was as low as 10 µg/mL [9].

Although the impregnation of cellulose paper with a layer of a responsive polymer may seem a straightforward task, interfacial incompatibility and poor delamination resistance are common drawbacks [14,15]. On the one hand, the convenience of using the probe as a dipstick for aqueous solutions requires the coating layer not to undergo elution, shrinkage, or excessive swelling by water. On the other hand, this implies that said polymer displays hydrophobic interactions, hindering a favorable attachment to the hydroxyl groups of cellulose. In the case of thermoplastic polymers with good film-forming properties, firm lamination can be attained by melting and spraying on a paper [16]. Still, dip coating is a much more common method when it comes to paper indicators such as pH paper and starch-iodide paper [17].

Considering the paper’s ability to wick water, it is easy to see why most proposals, not only for nitroaromatics sensing in particular but for paper-based indicators in general, assess the detection of the analyte in the solution phase. In light of this, we aim to show that the possibilities of paper-based probes can be extended to vapors from nitroaromatic compounds. It is known that TNT-filled land mines emit, besides TNT itself, the more volatile 2,4-dinitrotoluene (DNT) and dinitrobenzene (DNB) isomers [18]. Pablos et al. [10] quantified the response of copolymers containing primary and secondary amine groups to TNT vapors at 60 °C, but, regarding a copolymer with 2-(dimethylamino)ethyl methacrylate (DMAEMA) units, they stated that it “was insensitive to these vapors”. We show that if this very same monomer is the only repeating unit, and possibly due to the use of a mineral-filled cellulosic substrate as well, the resulting homopolymer is sensitive to TNT vapors.

For selectively sensing TNT vapors, this work opts for the polymerization of DMAEMA, as above hinted. For DNT and 1,2-dinitrobenzene (*o*-DNB) vapors, we choose fluorene and fluorene/phenylene-based copolymers (Figure 1), as their fluorescence is known to be quenched by nitroaromatics [1,6]. Unlike DMAEMA, fluorene and phenylene are highly hydrophobic monomers [19,20,21]. Hence, the polyfluorenes of choice in this work possess quaternary ammonium groups to promote compatibility with paper. Due to the remaining glucuronic units of hemicellulose and the polarization of hydroxyl groups, the surface of cellulose fibers has a certain negative charge [22,23]. All considered, we use polyDMAEMA, poly(9,9-bis(6′-*N*,*N*,*N*-trimethylammonium)hexyl)-fluorene phenylene) bromide (HTMA-PFP) and poly(9,9-bis(3′-(*N*,*N*-dimethyl)-*N*-ethylammoinium-propyl-2,7-fluorene)-alt-2,7-(9,9-dioctylfluorene))dibromide (PEDMA-DO-PBF), to impregnate paper strips, and we discuss the mechanisms involved and the conditions required for the effective detection of vapors from TNT, DNT, and *o*-DNB.

## 2. Results

PolyDMAEMA-coated paper strips progressively turned reddish or brown in the presence of TNT vapors but did not display any color change when exposed to DNT or *o*-DNB, even if a concentrated solution (1 mol/L) of them was painted on the strips. On the other hand, polyfluorene-coated paper strips were quenched, albeit not selectively, by nitroaromatics. Figure 2 shows one example of polyDMAEMA coloring with TNT under natural light, while the cerulean emission of poly(9,9-bis(6′-*N*,*N*,*N*-trimethylammonium)hexyl)-fluorene phenylene) bromide (HTMA-PFP) under UVA radiation was nearly suppressed by DNT. The paper strips in Figure 2a correspond to increasing exposure time to TNT at 60 °C. Those of Figure 2b were obtained after 18 h of exposure under different conditions: solid to vapor at 23 °C (*DNT_s_*, 23 °C), solid to vapor at 60 °C (*DNT_s_*, 60 °C), and evaporation from the solution phase at 60 °C (*DNT/ACN*), using acetonitrile as solvent. The mass of the nitroaromatic compound was always 25 mg, the volume of the vapor chamber was always the same (200 cm^3^), and only a small fraction of the analyte evaporated in all cases.

A more in-depth evaluation of this responsive behavior is discussed as follows. In order to attain a basic notion of the mechanism of fluorescence quenching of cationic polyfluorenes by nitroaromatics, this phenomenon was first evaluated in the solution phase.

### 2.1. Assessment of Fluorescence Quenching in Solution

Regardless of the quencher concentration, HTMA-PFP consistently had an absorption peak at 370 nm (Appendix A) and blue emission at 403 nm when subjected to UVA radiation. No bathochromic or hypsochromic shifts of these bands were detected by adding DNT (Figure 3). However, the two evident effects were fluorescence quenching and the increase of the relative intensity of the shoulder at 416 nm, to the point that it surpasses the peak at 403 nm for the highest concentration of DNT (800 ppm). Indeed, this ratio was used in a previous work as an indicator of the DNA-mediated quenching of HTMA-PFP [24]. In contrast, the ratio of the emission peak of PEDMA-DO-PBF (428 nm) to the second peak at 453 nm did not change significantly upon the addition of quencher. The maximum absorption wavelength of PEDMA-DO-PBF is 413 nm (Appendix A), located in the visible region, and thus not even a UVA lamp is strictly necessary.

The quenching efficiency (*QE*) that is displayed in Figure 3 was calculated from Equation (1):(1)QE%=100I0−II0
where *I_0_* is the highest emission intensity of the polyfluorene sample without nitroaromatics, and *I* is the highest emission intensity of each sample containing DNT or *o*-DNB. Likewise, *I_0_*/*I* can be plotted against the concentration of the quencher, observing certain linearity. The compliance of the steady-state fluorescence quenching to the Stern-Volmer relationship can thus be assessed [25]:(2a)I0I=1+KSVQ
where *K_SV_* is the Stern-Volmer constant and [*Q*] is the quencher concentration. It should be noted that no significant difference was observed in electronic absorption spectra upon the addition of DNT (Appendix A), and thus the ratio of emission intensities can be assumed to equal the ratio of quantum efficiencies. Due to the presence of different quenching regions (see Figure 3a), the modified Stern-Volmer equation [26]
(2b)I0I0−I=1fa+1faKaQ
where *f_a_* is the fraction of the initial fluorescence that is accessible to the quencher and *K_a_* is the Stern-Volmer quenching constant of the accessible fraction, has also been applied. The correlation coefficient is *R^2^* = 0.982 if the highest concentration is excluded.

It is generally assumed that fluorescence quenching is mainly mediated by the formation of an adduct in which the nitroaromatic compound acts as the electron acceptor [1,27]. If *QE* is the subsequent response of the fraction of polyfluorene that becomes bound to DNT or *o*-DNB, it is reasonable to fit its values to the Hill equation:(3)QE%=1001+QC50Qn
where *QC_50_* is the concentration of quencher that produces half the maximum response possible, and *n* is the Hill coefficient. These values and the Stern-Volmer constant in each case are displayed in Table 1. Overall, HTMA-PFP could be more easily quenched by low concentrations of nitroaromatics than its bisfluorene-based counterpart. This is discussed below in terms of whether the emission spectrum of the polymer overlaps the absorption spectrum of the quencher. It can also be seen that for DNT concentrations ranging from 200 to 600 ppm, a downward deviation from the Stern-Volmer equation (Equation (2a)) is observed. We can hypothesize that two different fluorophore populations [28] might be present as a consequence of the formation of aggregates in solution [29]. Hence, Figure 3a shows other two separate fittings to the Stern-Volmer plot (in green). Only in the case of HTMA-PFP with DNT, PET could be significantly complemented with RET in terms of contribution to the reduction of the emission intensity.

### 2.2. Performance of Polyfluorene-Coated Paper Strips

The most important difference that the paper format attains when compared to solutions, ultimately affecting the instructions that the user of the paper probes should follow, is a shift towards lower excitation wavelengths (Appendix A). The broad character of these spectra, instead of presenting narrow peaks, makes any cheap UVA lamp still suitable for the detection of the emission, but HTMA-PFP had its maximum excitation intensity at 349 nm (*Vs*. 370 nm in solution), while the maximum absorption of PEDMA-DO-PBF took place at 370 nm, although with a local maximum at 403 nm (*Vs*. 413 nm in solution). More precisely, solvation in aqueous systems exerts a hypsochromic effect, while evaporating the solvent promotes the opposite shift.

Emission suffered, however, a red shift. The maximum emission of HTMA-PFP was shifted from 403 nm in solution to 420 nm on paper. The first peak of PEDMA-DO-PBF was shifted from 428 nm in solution to 451 nm on paper. The second peak was both enhanced and shifted towards the green region in both cases, as can be seen from the spectra in Figure 4. This graph also quantifies the *QE* for each set of conditions tested. The positive influence of temperature is not attributed to a more favored formation of quenched adducts, but simply to the great increase of the vapor pressure. In the case of DNT, it goes from 2.77 × 10^−7^ bar at 23 °C (277 ppbv) to 1.7 × 10^−5^ bar at 60 °C (15 ppmv, assuming compliance to Gay-Lussac’s law). These values were estimated from the Aspen Plus package [30].

Along with the broadening of emission bands, the red shifts in emission implied that HTMA-PFP-coated papers appeared azure or cerulean, instead of violet-blue, under UVA radiation (Table 2). PEDMA-DO-PBF-coated papers could be seen as cornflower blue.

Lacking a fluorimeter and employing a UVA lamp, the user could visually detect quenching, as shown in Figure 2b, and therefore know qualitatively if there are nitroarene vapors or not. Besides, by incorporating image analysis software or even one of the many smartphone apps for RGB mapping, such as *Colorimetric Titration*, it is possible to quantify it by the decrease in the blue and green values. The response can be unified in only one magnitude, the modulus of the {Δ***G***,Δ***B***} vector:(4)$$\left|\Delta GB\right|=\sqrt{{\left(-\Delta G\right)}^{2}+{\left(-\Delta B\right)}^{2}}$$
where Δ***G*** and Δ***B*** are the increments in green and blue emission, thus being negative when it comes to quenching. The resulting mean values are shown in Table 2, along with a typical paper strength parameter. The machine-direction breaking length decreased by dip coating with polyfluorene/SDS dispersions, from 9.7 ± 0.2 km to 6.2 ± 0.2 km. By separating adjacent fibers, surfactants weaken inter-fiber bonding, thus lowering the dry strength of the paper [31]. Still, there is no doubt that the impregnated paper is strong enough to be used as a probe for vapors or as a dipstick for liquids.

### 2.3. Performance of polyDMAEMA-Coated Paper Strips

The color of TNT-sensing paper was blush, red barn, or maroon after exposure, given that the Meisenheimer complex absorbs light in the blue and green regions. Therefore, the parameter |Δ***GB***| is still useful to assess the evolution of the visual response to the exposure to vapors. Indeed, while the turn-off of blue/green emission means that the 8-bit values of those colors decrease, acquiring red-brownish shades can be reflected by negative Δ***G*** and Δ***B*** as well. Hence, Figure 5 expresses the modulus of the Δ***GB*** two-dimensional vector, calculated from Equation (4), as a function of the exposure time at 60 °C. At this temperature, the vapor pressure of TNT is 8.95 × 10^−7^ bar, equivalent to an ideal gas concentration of 790 ppbv.

Moreover, the colorimetric or RGB-based absorbance (*A*) could be estimated from the blue coordinate [32]:(5)$$A_{B}-A_{B,0}=-\gamma \;log\frac{B}{B_{0}}$$
where the subindex 0 refers to the blank and *γ* is the gamma correction value, which will be assumed to be 1. Working with solids, we cannot expect this estimated absorbance to be directly proportional to the actual amount of Meisenheimer complex at the interface, but it is reasonable to imply that there is some monotonously increasing relationship. A power law could then describe the evolution with time with a correlation coefficient (R^2^) of 0.9993:(6)$$A_{B}-A_{B,0}=0.107\;t^{0.44}$$
where *t* is the exposure time. With all due caution, it can be hypothesized that kinetics is governed by limitations to mass transfer, particularly diffusion. This is discussed below.

Although heating requirements limit usefulness, the alternative is to expose the sample to the paper strip for unfeasibly long times. A timid response (Δ***GB*** = 27) could also be appreciated at 23 °C after 5 days, but this is even lower than the increment after only 2 h at 60 °C. At room temperature, the solid vapor pressure of TNT is as low as 9.16 × 10^−8^ bar, corresponding to 9.16 ppbv under the assumption of ideal gas.

It is worth mentioning that, when paper-based probes were used as dipsticks in a TNT solution, instead of being exposed to vapors, the color change was fast, as observed by Pablos et al. [10] for polyDMAEMA-coated fabrics.

Like the impregnation of paper strips with surfactants, dip coating with polyDMAEMA aqueous-alcoholic solution also exerted a loss of paper strength but was less dramatic. The breaking length of the paper in the machine direction went from 9.7 to (7.0 ± 0.3) km. The mechanism of paper weakening by wetting with alcohols is different from that by wetting with surfactant solutions. When drying, the presence of ethanol promotes the evaporation of so much water along with it, suppressing fiber-water-fiber H-bonding interactions that contributed to paper strength. Once again, in any case, the dry strength of the resulting strips was more than enough for this application.

## 3. Discussion

Overall, the responsive polymers chosen in this study were proven useful for the visual and/or software-assisted detection of nitroarmatic vapors, but important limitations should be mentioned. Both for polyfluorenes and polyDMAEMA, the key drawback is not related to a limit of detection, but to the long times required by low concentrations of vapors from nitroaromatic explosives to yield a noticeable visual response. The formation of an adduct could be expressed in a simple way as an equilibrium:(7)$$ED^{\cdot }{}_{\left(s\right)}+EA_{\left(g\right)}⇌{\left[ED\cdot EA\right]}_{\left(s\right)}$$
where *ED* is the electron donor (polyfluorenes, polyDMAEMA) and *EA* is the electron acceptor (nitroaromatics). The limitation does not lie in the kinetics of Equation (7), as electron transfer is generally faster than diffusion through polymers. There is a plausible explanation both for the need for many hours and for the visualization of a response even in very dilute atmospheres, though. As the nitroaromatic compound becomes attached, as an electron acceptor, to an >N– group or to fluorene/phenylene units on paper, the equilibrium shifts to the right due to depletion of the only substance that is in the gas phase. The response is thus triggered by the accumulation of immobilized nitroaromatics beyond the concentration indicated by their vapor pressure.

Moreover, considering the poor response where DNT was evaporated from acetonitrile, despite its partial vapor pressure being higher than the solid vapor pressure, these paper probes, if intended for vapors, should be used without dissolving nitroaromatics first. Most likely, this negative interference of acetonitrile would not have been avoided by choosing any other water-miscible organic solvent. Virtually, all conventional choices behave as Lewis bases in the presence of a strong electron acceptor. The heat of binding of acetonitrile to BF_3_ (a paradigmatic Lewis acid) is 60 kJ/mol or 32 kJ/mol in vapor phase [33], but those of ethanol, acetone, tetrahydrofuran, pyridine, dimethylsulfoxide (DMSO), DMF and dimethylacetamide are even higher [34].

### 3.1. Insights into Fluorescence Quenching

Generally, the fluorescence quenching of conjugated polymers by nitroaromatics is explained by a photoinduced electron transfer (PET) phenomenon [1,2,27,35]. While nitroarenes are strong electron acceptors, the aromatic nucleus of polyfluorenes would act as donors. In this work, PET is proposed as the primary mechanism in the case of PEDMA-DO-PBF quenching by *o*-DNB, since quenching occurs despite no overlap between the fluorescence emission spectrum of the donor and the absorption spectrum of the acceptor.

Nonetheless, some reasons compel us to suggest that for HTMA-PFP quenching by DNT, neither PET nor resonance energy transfer (RET) should be neglected. First, alkyl chains with quaternary ammonium groups (Figure 1) are deactivating substituents that diminish the electron-donating capacity, which could result in PET not outweighing other causes for quenching. Secondly, DNT, a yellow-orange compound, absorbs blue light, while HTMA-PFP (under photoexcitation) emits, as depicted in Figure 2b, blue light, i.e., the overlap condition for RET is fulfilled [36]. This is highlighted by the fact that DNT exerts a more marked quenching effect on HTMA-PFP’s emission peak at 403 nm, where the absorbance of DNT is higher than at 416 nm, corresponding to the shoulder in the emission spectrum of HTMA-PFP (Figure 3a).

The partially crippled electron-donating capacity of cationic polyfluorenes also explains the relatively low Stern-Volmer constants (Table 1). For neutral conjugated polymers, such as 2,1,3-benzooxadiazole-alt-polyfluorene, values of 6.9 × 10^3^ M^−1^ have been reported in quenching assays with DNT [37].

By fitting the data on the HTMA-PFP/DNT system to the modified Stern-Volmer equation (Equation (2b)), we can conclude that only 15% of the fluorophore is not available to interact with the quencher. The system does not comply with the linear plot unless two sets of data points (up to 100 ppm and 200–600 ppm) are distinguished. Finally, at the highest DNT concentration, the quenching is again enhanced, which can be justified by the quenching sphere of action [28], as it was also found by Dong et al. [13].

### 3.2. Interactions Involving the Substrate, the Responsive Polymers, and the Surfactant

Even if the requirement for relatively high concentrations of nitroaromatics to exert noticeable quenching is a disadvantage in terms of the limits of detection, quaternary ammonium groups are undoubtedly convenient. They ease its disaggregation in the polar systems that paper coating requires, also facilitating its binding to the negatively-charged paper surface. No less importantly, it has been shown that the interaction with SDS, at the critical micelle concentration or above, notoriously boosts the fluorescence of the conjugated macromolecule [38,39].

It is worth remarking that the surface charge of the cellulosic substrate is, albeit slight, negative [21]. The quaternary ammonium groups from cationic polyfluorenes are able to form ion-dipole bonds with the polarized hydroxyl groups of cellulose. Furthermore, the substrate in this work differs from most paper-based probes reported, which usually use filter paper, in that its properties are more similar to those of printing paper, but without brightening agents. Mineral fillers provide a more negatively-charged surface when immersed in aqueous systems, shift the hydrophilic/hydrophobic balance towards the hydrophobic side (hindering elution by water), and decrease the porosity [40], allowing to concentrate the visual response on the surface.

Another reason why positively charged polyfluorenes was chosen in this work is that quaternary ammonium groups make the fluorescence of the polymer insensitive to cationic quenchers, increasing the selectivity towards neutral electron acceptors such as nitroaromatics [1], although the possible interference from anionic quenchers such as DNA must be taken into account [24]. It should be noted that, in spite of the charged hydrophilic groups, none of these polymers can be dissolved in water in lieu of a surfactant or an organic co-solvent. The presence of alkyl chains in the positively-charged polyfluorenes is key to avoiding dissolution in water and, therefore, elution after dip coating. Moreover, besides providing hydrophobicity, they grant longer fluorescence lifetimes [41]. Furthermore, functionalization of the fluorene units in the C-9 position is known to improve their photostability [42].

As for the choice of chromogenic polymer for the naked-eye detection of TNT, polyDMAEMA, both tertiary amines, and carbonyl groups are hydrogen bond acceptors. Since the structure of this polymer does not contain H-donor groups, this kind of intermolecular interaction will only be established with water and the hydroxyl groups of cellulose. The presence of mineral fillers, in another context, helps hinder the diffusion of the polymer through the matrix. This, along with the choice of a polymer entirely consisting of DMAEMA units, avoided the dilution of the visual response and allowed for an effective polyDMAEMA-paper system for the selective detection of TNT.

### 3.3. Concluding Remarks

Although detecting nitroaromatics in a water-miscible, volatile organic solvent is undoubtedly useful for groundwater pollution monitoring and soil recovery, e.g., as in the assessment of the extracts from soil flushing, we recommend the use of the paper strips reported here as direct dipsticks in that case. Dipping polyDMAEMA-coated fabrics into TNT solutions was shown to trigger coloring with a limit of detection of 1.1 × 10^−4^ M [3].

In conclusion, the progressive accumulation of evaporated nitroaromatic by chemisorption on a functionalized paper strip shifts the key for detection from concentration to the exposure time. While the need for high temperature and/or several hours of exposure hinders the user-friendliness of these probes, this problem is surmountable, to our understanding, by future research. For instance, attaining a rougher surface by changing the coating technique would ease sub-surface lateral adsorption and decrease the limitations to diffusion.

## 4. Materials and Methods

### 4.1. Reagents, Analytes, and Paper

DMAEMA (98%) was purchased from Sigma-Aldrich (Lyon, France). Its polymerization towards polyDMAEMA was performed with azo-bis-isobutyronitrile and in a polar aprotic medium (*N*,*N’*-dimethylformamide, DMF), as described and characterized in previous work from Pablos et al. [10]. HTMA-PFP was a kind gift from Dr. Ricardo Mallavia from University Miguel Hernandez, Spain, and it was synthesized by a Suzuki coupling reaction from 1,4-phenyldiboronic acid and 2,7-dibromo-9,9-bis(6′-bromohexyl)fluorine [24]. The other polyfluorene involved in this work PEDMA-DO-PBF, is commercially available by the name “PFN-Br”. It was received from Sigma-Aldrich (Lyon, France).

Regarding the nitroaromatic compounds to be used as analytes, both DNT and *o*-DNB were purchased from Sigma-Aldrich (Lyon, France). TNT was synthesized from DNT as follows: 2 g of DNT were dissolved in 10 mL of fuming sulphuric acid in a 3-necked round bottom flask equipped with a condenser and a thermometer. The mixture was heated at 75 °C, and a mixture of 10 mL of fuming sulphuric acid and 10 mL of fuming nitric acid was added dropwise. Simultaneously to the addition, the temperature was increased stepwise up to 96 °C, and then slowly to 120 °C for 30 min. After 30 min at 120 °C, the reaction medium was cooled to room temperature. The mixture was poured into hot water (85–90 °C). The yellowish solid was filtered and washed twice with hot water and sodium bicarbonate. Then, ^1^H and ^13^C NMR spectra were recorded by means of an Avance III HD spectrometer (Bruker Corp., Billerica, MA, USA). Yield: 52%; ^1^H NMR (300 MHz, DMSO-d_6_) δ(ppm) = 9.03 (s, 2H), 2.57 (s, 3H); ^13^C NMR (75 MHz, DMSO-d_6_) δ(ppm) = 151.31, 146.16, 133.42, 123.03, 15.38 (Appendix A).

All coating experiments were performed on uncoated paper sheets of industrial origin, possessing a basis weight of 90 g m^−2^. This paper proceeded from bleached kraft wood pulp, it was uncoated but calendered, and it possessed no optical brightening agents, but it contained paper-grade mineral fillers of undisclosed nature. Its Gurley air permeability was determined as 18 ± 1 s, its Bendtsen roughness was 86 ± 10 mL/min, and its breaking length was 9.7 ± 0.2 km.

### 4.2. Dip Coating

As DMF is not a proper solvent for paper coating, the volume of the polyDMAEMA solution was reduced by codistillation with diethyl ether and the polymer was precipitated by adding water. After recovery and drying, 1 g of polyDMAEMA was easily dissolved in 100 mL of a water-ethanol (1:9, *v*/*v*) mixture.

Polyfluorene-based formulations were prepared by suspending 0.4 g of each of the conjugated polyelectrolytes in 100 mL of water, in presence of 0.8 mmol of sodium dodecylsulfate (SDS). The mixture was stirred at room temperature for 24 h. The resulting solution did not settle when undisturbed. All experiments involving each of the conjugated polymers was performed from the same polyfluorene/SDS/water stock solution.

Once all coating formulations were prepared, impregnation was performed by means of a KSV Nima Dip Coater, for 1 min. Coated paper strips were subsequently dried for 5 min by a thermoventilator at 50–60 °C.

### 4.3. Exposure to Nitroaromatics

Each of the responsive polymers was assigned to one of the nitroaromatic compounds. PolyDMAEMA was used to test the presence of TNT for obvious reasons since it does not form Meisenheimer complexes with DNT or *o*-DNB. The former (the weakest electron acceptor of the three) was tested with HTMA-PFP, while the latter was assessed by PEDMA-DO-PBF.

Every coated paper strip was exposed to vapors from the corresponding nitroaromatic compound in a similar way. The analyte (25 mg) was deposited at the bottom of a vapor chamber. The functionalized strip was left to hang from the top and the chamber was sealed. Then, this closed vessel was either kept at room temperature (23 °C) or placed in an oven at 60 °C for certain time lapses. In the latter case, the nitroaromatic compound and the chamber were briefly preheated to avoid the influence of heat transfer limitations.

Before spectrofluorometry or photography analysis, paper strips were allowed to cool down in an exicator overnight. Meanwhile, they were protected from light with aluminum foil until characterized.

### 4.4. Spectrofluorometry

Polyfluorene dispersions were diluted to 5 ppm, SDS was added to readjust its concentration to 8 mmol/L, and the luminescence of each of the conjugated polymers was quenched with increasing concentrations of nitroaromatics. The fluorescence spectra of solutions were recorded on a Jovin-Yvon Spex Fluorog 3-2.2 spectrofluorometer, set with a right-angle configuration, slit width of 1 nm, and an integration time of 1 s. The light source was a 450 W ozone-free xenon arc lamp [43].

Regarding the fluorescence spectra of paper samples, the same device was used in a front-face (45°) configuration, slit width of 0.5 nm, and an integration time of 1 s. Excitation spectra were recorded from 240 to 400 nm. Emission spectra were recorded with UVA excitation and ranging from 340 to 520 nm.

### 4.5. Evaluation of the Usability of Paper-Based Probes

Pictures were taken with a conventional smartphone, either under white light and over a white background for polyDMAEMA-coated papers, or under UV radiation and over a black background for polyfluorene-coated papers. These photographs include the coated paper strips right after drying and identical strips that have been exposed to nitroaromatics. The RGB24 coordinates of the color of each strip were computed by means of Adobe Photoshop’s color picking tool. This software pack was also used to exert the minimum brightness adjustment of the whole picture so that at least one of the coordinates of the RGB vector for the uncoated paper was 255, in the case of natural light, or 0 in the case of UV radiation.

To ensure that coating and drying did not weaken paper excessively, the breaking length was measured by means of a universal testing machine, following the ISO standard 1924 [44].

## Figures and Tables

**Figure 1 molecules-27-02900-f001:**
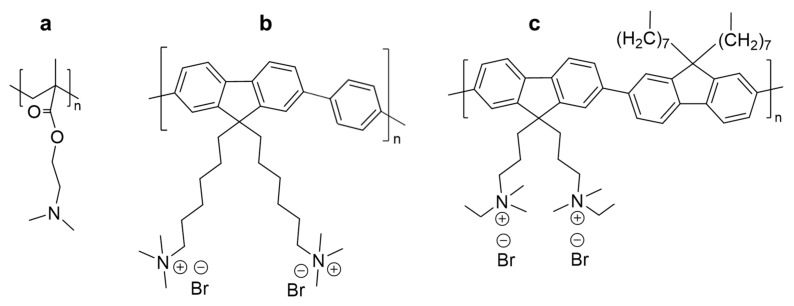
Chemical structures of the responsive polymers involved in formulations for functionalizing paper: (**a**) poly(2-(dimethylamino)ethyl methacrylate)—polyDMAEMA; (**b**) poly(9,9-bis(6′-*N*,*N*,*N*-trimethylammonium)hexyl)-fluorene phenylene) bromide—HTMA-PFP; (**c**) poly(9,9-bis(3′-(*N*,*N*-dimethyl)-*N*-ethylammoinium-propyl-2,7-fluorene)-alt-2,7-(9,9-dioctylfluorene))dibromide—PEDMA-DO-PBF.

**Figure 2 molecules-27-02900-f002:**
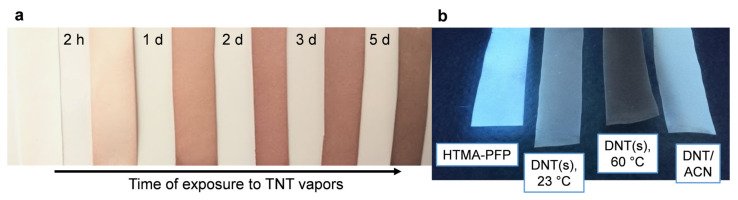
Photographs of: (**a**) PDMAEMA-impregnated paper for TNT, the first of which was not in contact with the analyte, whereas each of the other samples has its exposure time indicated at its left; (**b**) HTMA-PFP-impregnated paper-based probes for DNT under radiation at 366 nm, where the sample at the left had not been in contact with the analyte.

**Figure 3 molecules-27-02900-f003:**
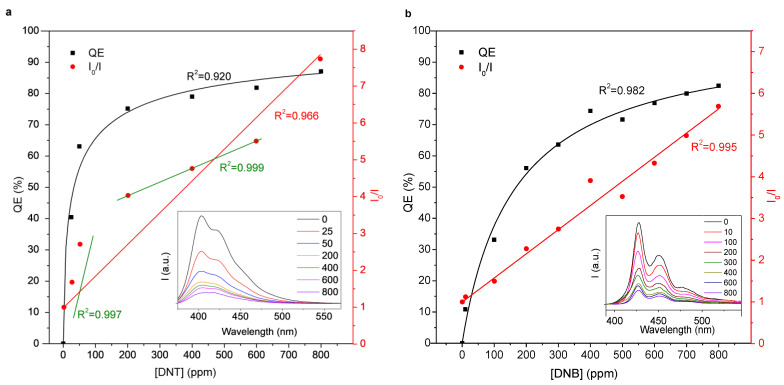
Fluorescence quenching of HTMA-PFP (**a**) and PEDMA-DO-PBF (**b**), in solution, with DNT and *o*-DNB, respectively, showing both linear fits to *I_0_*/*I* (Stern-Volmer plot) and a hyperbolic fit to *QE* (black), corresponding to the Hill equation. The inset figure displays the emission spectra, under excitation at 375 nm (**a**) or 385 nm (**b**). Legend numbers are expressed in ppm.

**Figure 4 molecules-27-02900-f004:**
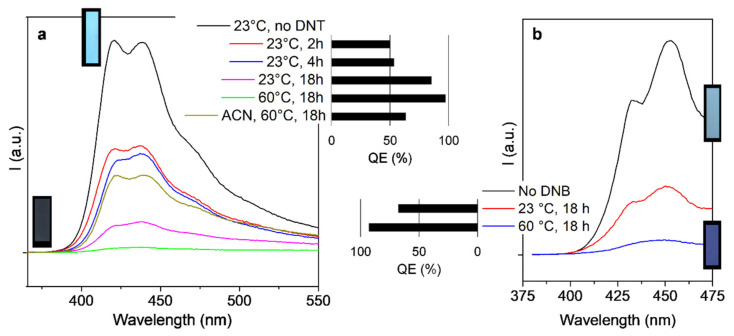
Fluorescence quenching of paper strips coated with HTMA-PFP (**a**) and PEDMA-DO-PBF (**b**), with DNT and *o*-DNB, respectively. The quenching efficiency is represented as a 0–100% bar in each case. Pictures for the extreme cases (no nitroaromatics and maximum *QE*) under UVA radiation are provided.

**Figure 5 molecules-27-02900-f005:**
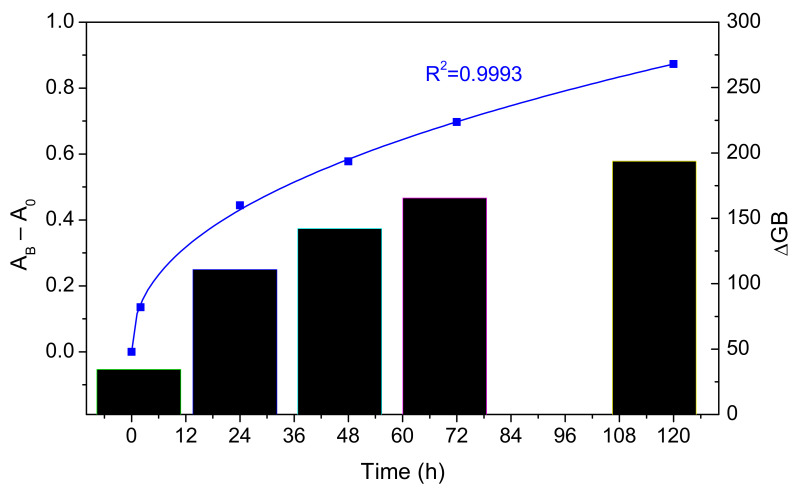
Evolution of the color change (black columns) and of the colorimetric absorbance (blue dots and line) with the time of exposure of polyDMAEMA-impregnated paper sheets to TNT.

**Table 1 molecules-27-02900-t001:** Fitting parameters for the fluorescence quenching of cationic polyfluorenes with nitroaromatics.

System	HTMA-PFP + DNT	PEDMA-DO-PBF + DNB
*K_SV_* (M^−1^)	(6.0 ± 0.4) × 10^3^	(1.05 ± 0.03) × 10^3^
*K_a_* (M^−1^), *f*_a_	(7 ± 1) × 10^3^, 0.85 ± 0.05	
Hill coefficient, *n*	0.58 ± 0.08	0.97 ± 0.08
*QC*_50_ (ppm)	33 ± 8	170 ± 10

**Table 2 molecules-27-02900-t002:** Color properties and strength of the fluorescent paper-based probes, highlighting the loss of green and blue emissions by exposure to nitroaromatics.

System	Uncoated	HTMA-PFP	PEDMA-DO-PBF
Excitation wavelength	-	ca. 349 nm	ca. 370 nm
Emission color	-	Light cerulean	Cornflower blue
|Δ***GB***| at 23 °C, 18 h	-	125	53
|Δ***GB***| at 60 °C, 18 h	-	243	116
Breaking length (km)	9.7 ± 0.2	6.2 ± 0.2	

## Data Availability

The CSV files corresponding to emission and absorption spectra will be made publicly available in the largest Portuguese data repository, https://www.rcaap.pt/ (accessed on 7 April 2022).

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
