# Peer review of "Paper-Based Probes with Visual Response to Vapors from Nitroaromatic Explosives: Polyfluorenes and Tertiary Amines"

_molecules, 2022, doi:10.3390/molecules27092900_

Round 1

Reviewer 1 Report

Aguado and his co-workers present in this manuscript the synthesis of three polymers, namely polyDMAEMA, poly(9,9-bis(6′-N,N,N-trimethylammonium)hexyl)-fluorene phenylene) bromide (HTMA-PFP) and poly(9,9-bis(3’-(N,N-dimethyl)-N-ethylammoinium-propyl-2,7-fluorene)-alt-2,7-(9,9-dioctylfluorene))dibromide (PEDMA-DO-PBF), the impregnation of selected industrial paper strips with these polymers, and the application of these impregnated papers for the detection of vapors from TNT, DNT and o-DNB. The mechanisms of vapor detection and the required conditions have also been discussed. In general, the research is carefully done and the paper is well-written. The content of the manuscript fits the scope of the journal.

Comments:

--- Figure 3a: the relationship between DNT concentration and I0/I is obviously not linear; authors should comment on this, as well as on the limitations of Stern-Volmer equation;

--- line 353-355: please provide reference for the synthesis of TNT, or a detailed description of the synthesis with proper structural characterization;

--- lines 356-357: please provide more details about the uncoated paper sheets of industrial origin, as this may be one of the key factors of reproducing this work.

Author Response

Aguado and his co-workers present in this manuscript the synthesis of three polymers, namely polyDMAEMA, poly(9,9-bis(6′-N,N,N-trimethylammonium)hexyl)-fluorene phenylene) bromide (HTMA-PFP) and poly(9,9-bis(3’-(N,N-dimethyl)-N-ethylammoinium-propyl-2,7-fluorene)-alt-2,7-(9,9-dioctylfluorene))dibromide (PEDMA-DO-PBF), the impregnation of selected industrial paper strips with these polymers, and the application of these impregnated papers for the detection of vapors from TNT, DNT and o-DNB. The mechanisms of vapor detection and the required conditions have also been discussed. In general, the research is carefully done and the paper is well-written. The content of the manuscript fits the scope of the journal.

We thank the reviewer for considering our manuscript and for the comments below, which undoubtedly helped us improve some key points.

Comments:

--- Figure 3a: the relationship between DNT concentration and I0/I is obviously not linear; authors should comment on this, as well as on the limitations of Stern-Volmer equation;

Kindly check this additions in the revised version:

Results: Due to the presence of different quenching regions (see Fig. 3a), the modified Stern-Volmer equation [26] (...) where fa is the fraction of the initial fluorescence that is accessible to the quencher and Ka is the Stern-Volmer quenching constant of the accessible fraction, has also been applied. The correlation coefficient is R2=0.982 if the highest concentration is excluded.

(...)

It can also be seen that for DNT concentrations ranging from 200 to 600 ppm, a downward deviation from the Stern-Volmer equation (Eq. 2a) is observed. We can hypothesize that two different fluorophore populations [28] might be present as a consequence of the formation of aggregates in solution [29]. Hence, Fig. 3a shows other two separate fittings to the Stern-Volmer plot (in green).

DiscussionIt can also be seen that for DNT concentrations ranging from 200 to 600 ppm, a downward deviation from the Stern-Volmer equation (Eq. 2a) is observed. However, by fitting these data to Eq. 2b, we can conclude that the Stern-Volmer equation is similar (taking into account the experimental error) and only 15% of the initial fluorescence is not available to interact with the DNT. Consequently, we can also hipothesise that two different fluorophore populations [26] might be present as a consequence of the formation of aggregates in solution [27]. Finally, at the highest DNT concentration, quenching is again enhanced, which can be justified by the quenching sphere of action [26], as it was also found by Dong et al. [28]. 

--- line 353-355: please provide reference for the synthesis of TNT, or a detailed description of the synthesis with proper structural characterization;

Addition:

4.1: TNT was synthesized from DNT as follows. 2 g of DNT were dissolved in 10 mL of fuming sulphuric acid in a 3-necked round bottom flask equipped with a condenser and a thermometer. The mixture was heated at 75 °C, and a mixture of 10 mL of fuming sulphuric acid and 10 mL of fuming nitric acid was added dropwise. Simultaneously to the addition, the temperature was increased stepwise up to 96 °C, and then slowly to 120 °C during 30 min. After 30 min at 120 °C, the reaction medium was cooled to room temperature. The mixture was poured into hot water (85-90 °C). The yellowish solid was filtered and washed twice with hot water and sodium bicarbonate. 1H and 13C NMR spectra were recorded by means of an Avance III HD spectrometer (Bruker Corp., Billerica, MA, USA).  Yield: 52%. 1H NMR (300 MHz, DMSO-d6) δ(ppm) = 9.03 (s, 2H), 2.57 (s, 3H). 13C NMR (75 MHz, DMSO-d6) δ(ppm) = 151.31, 146.16, 133.42, 123.03, 15.38 (Fig. S4).

Please see the attachment. It is now provided as supplementary information.

--- lines 356-357: please provide more details about the uncoated paper sheets of industrial origin, as this may be one of the key factors of reproducing this work.

Indeed, most works on paper-based detectors use filter paper (all-cellulose, ashless), and the presence of components other than cellulose may affect sorption significantly. We now provide its grammage, the fact of being kraft and bleached, and the fact of having mineral fillers but no organic brightening agents.

This paper is not commercially available, as most commercial white papers contain some sort of optical fluorescent agent that would cause interferences.

The fillers (although the manufacturer is opaque to us regarding the exact composition) hinder the diffusion of the responsive polymers through the "z" direction and ease the retention of hydrophobic macromolecules.

All coating experiments were performed on uncoated paper sheets of industrial origin, possessing a basis weight of 90 g m–2. This paper proceded from bleached kraft wood pulp, it was uncoated but calendered, and it possessed no optical brightening agents, but it contained paper-grade mineral fillers of undisclosed nature.

On one hand, this non-disclosure of fillers (most likely Ca/Si/Al minerals...) affects reproducibility. On the other hand, we still hold that "coating" filter paper would have been less defendible: it would not be a surface treatment, it would be less resistant to elution and it would have implied the dilution of expensive polyfluorenes through the whole paper matrix.

Reviewer 2 Report

Valente and co-workers present several coated paper substrates for detecting nitroaromatic explosives in vapour and in solution. Whilst the materials and methods used are well known, the efforts seem solid, and the topic is an important one.

I have a few comments that must be addressed before the work could be considered publishable however:

The abstract claims novelty for use of paper as a test strip substrate for detecting explosives. This is not novel – there are examples of this in the literature e.g. https://pubs.rsc.org/en/content/articlehtml/2018/ra/c7ra13536j as well as exmaples on fibres, in films and so on.

Also in the abstract the authors suggest that the issue of long-time-to-quench or requirement for high vapour pressures to achieve a signal might be overcome in future, but don’t suggest how this may be achievable.

On line 108 I think the second sample should be called “DNTs, 60 oC”?

It wasn’t clear to me if Figure 3 was using vapour phase materials or solutions? It was clear however that the I0/I data in (a) cannot be fitted with a straight line. The authors should revisit this data and try to explain what might be occurring here to obtain this curve. In general Figure 3 needed to be re aligned, have the same axes ranges for both plots etc. 

The text makes reference to a white dot in Figure 5, but the caption/legend of the figure do not – is it deltaGB or Ab-A0? I wasn’t convinced that this white dot showed any response beyond hour/day 0.

Unfortunately the supplementary materials were not provided for review so I could not assess these. 

Author Response

Valente and co-workers present several coated paper substrates for detecting nitroaromatic explosives in vapour and in solution. Whilst the materials and methods used are well known, the efforts seem solid, and the topic is an important one.

We deeply thank the reviewer for considering our manuscript and for their insightful comments.

I have a few comments that must be addressed before the work could be considered publishable however:

The abstract claims novelty for use of paper as a test strip substrate for detecting explosives. This is not novel – there are examples of this in the literature e.g. https://pubs.rsc.org/en/content/articlehtml/2018/ra/c7ra13536j as well as exmaples on fibres, in films and so on.

We never intended to implied that paper-for-explosives was novel but, admittedly, the abstract was perhaps too enthusiastic. Citing the introduction in the original submission:

"There have already been valuable advances in this direction. For example, in a highly cited work, Ma et al. reported the impregnation of filter paper with 8-hydroxyquinoline aluminum (Alq3)-based nanospheres, and their fluorescence was quenched by 2,4,6-trinitrotoluene (TNT) solutions [12]. As for the colorimetric approach, polyaniline-coated papers have also been found to turn reddish in the presence of TNT, forming a Meisenheimer complex, with visually noticeable changes even when the analyte concen-tration was as low as 10 µg/mL [9]"

Works on fabrics and even nanocellulose films were also cited. Still, as indicated ("the abstract"), the reviewer probably refers to "the potential of paper as a substrate for those macromolecules is yet to be explored", where the ambiguous "is yet to be explored" has now been replaced with "can be further developed".

Addition to the revised version, introduction:

"In an approach that is more similar to the one presented here, filter paper strips were drop-coated with polytriphenylamine derivatives that were effectively quenched by nitroaromatics in the micromolar range [13]."

Also in the abstract the authors suggest that the issue of long-time-to-quench or requirement for high vapour pressures to achieve a signal might be overcome in future, but don’t suggest how this may be achievable.

We have deleted this in the revised version, fearing that it was too vague. 

On line 108 I think the second sample should be called “DNTs, 60 oC”?

It has been fixed. We are grateful.

It wasn’t clear to me if Figure 3 was using vapour phase materials or solutions? It was clear however that the I0/I data in (a) cannot be fitted with a straight line. The authors should revisit this data and try to explain what might be occurring here to obtain this curve. In general Figure 3 needed to be re aligned, have the same axes ranges for both plots etc.

Figure 3 has been changed and the phase is clarified. The non-linearity has also been extensively discussed, including additional fittings:

Results: Due to the presence of different quenching regions (see Fig. 3a), the modified Stern-Volmer equation [26] (...) where fa is the fraction of the initial fluorescence that is accessible to the quencher and Ka is the Stern-Volmer quenching constant of the accessible fraction, has also been applied. The correlation coefficient is R2=0.982 if the highest concentration is excluded.

(...)

It can also be seen that for DNT concentrations ranging from 200 to 600 ppm, a downward deviation from the Stern-Volmer equation (Eq. 2a) is observed. We can hypothesize that two different fluorophore populations [28] might be present as a consequence of the formation of aggregates in solution [29]. Hence, Fig. 3a shows other two separate fittings to the Stern-Volmer plot (in green).

Discussion: It can also be seen that for DNT concentrations ranging from 200 to 600 ppm, a downward deviation from the Stern-Volmer equation (Eq. 2a) is observed. However, by fitting these data to Eq. 2b, we can conclude that the Stern-Volmer equation is similar (taking into account the experimental error) and only 15% of the initial fluorescence is not available to interact with the DNT. Consequently, we can also hipothesise that two different fluorophore populations [26] might be present as a consequence of the formation of aggregates in solution [27]. Finally, at the highest DNT concentration, quenching is again enhanced, which can be justified by the quenching sphere of action [26], as it was also found by Dong et al. [28].

The text makes reference to a white dot in Figure 5, but the caption/legend of the figure do not – is it deltaGB or Ab-A0? I wasn’t convinced that this white dot showed any response beyond hour/day 0.

We did not want to imply that the response appears "all of a sudden" after 5 d at room temperature, but even after this time it is very timid — even lower than that obtained after only 2 h at 60 °C. In order not to mislead the reader, the dot has been removed and the statement that RT detection could be unfeasibly long but possible (as long as the system is closed to mass transfer) is relegated to the text.

Unfortunately the supplementary materials were not provided for review so I could not assess these.

Attached.

Round 2

Reviewer 3 Report

The manuscript has been substantially improved. It is now suitable for publication